# Toward Human Readable Prompt Tuning:
# Kubrick's *The Shining* is a good movie, and a good prompt too?

**Weijia Shi**[*]     **Xiaochuang Han**[*]
**Hila Gonen    Ari Holtzman    Yulia Tsvetkov    Luke Zettlemoyer**
Paul G. Allen School of Computer Science & Engineering,
University of Washington, Seattle, WA
{swj0419, xhan77, hilagnn, ahai, yuliats, lsz}@cs.washington.edu

## Abstract

Large language models can perform downstream tasks in a zero-shot fashion, given natural language prompts that specify the desired behavior. Such prompts are typically hand engineered, but can also be learned with gradient-based methods from labeled data. However, it is underexplored *what factors* make the prompts effective, especially when the prompts are in natural language. In this paper, we investigate common attributes shared by effective prompts in classification problems. We first propose a human readable prompt tuning method (FLUENTPROMPT) based on Langevin dynamics that incorporates a fluency constraint to find a distribution of effective and fluent prompts. Our analysis reveals that *effective prompts are topically related to the task domain* and *calibrate the prior probability of output labels*. Based on these findings, we also propose a method for generating prompts using only unlabeled data, outperforming strong baselines by an average of 7.0% accuracy across three tasks. We release our code and data in [github.com/swj0419/FluentPrompt](github.com/swj0419/FluentPrompt).

## 1  Introduction

Large language models (LMs) can perform downstream tasks by simply conditioning on a prompt–a short sequence of text specific to the task. Such natural language prompts are either carefully hand engineered (e.g., manual prompt engineering, Kojima et al. 2022) or automatically learned from labeled data (e.g., gradient-based prompt tuning, Shin et al. 2020). Despite their effectiveness, it remains unclear what makes these prompts work and what attributes effective prompts share. In this paper, we aim to identify key characteristics of effective prompting, and use this knowledge to generate effective and human readable prompts without any labeled data.

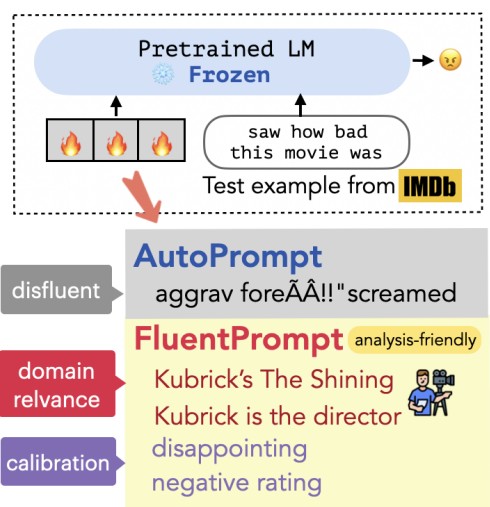

Figure 1: Compared with previous discrete prompt tuning method AutoPrompt (Shin et al., 2020) which generates gibberish prompts, FLUENTPROMPT can identify effective and more readable prompts, useful for downstream analyses. We find that in prompt tuning, the good prompts **are topically relevant to the task domain** (e.g., mentioning a film director "Kubrick" in a movie sentiment classification task), and **calibrate the prior probability of output labels** (e.g., including negative words to balance an overly optimistic model).

There are two main challenges for performing this type of analysis. First, manual prompt tuning produces a limited number of effective prompts for each task, making it difficult to infer common features of good prompts where contrast with less effective prompts is needed. On the other hand, the prompts found by gradient-based tuning methods are often disfluent and unnatural, making them difficult to interpret (e.g., AutoPrompt in Figure 1).

To overcome these challenges, we first propose a human readable prompt tuning method called FLUENTPROMPT. Inspired by prior work in controllable text generation (Kumar et al., 2022), FLU-ENTPROMPT uses Langevin dynamics to generate a set of human readable prompts for any task. Our

---

[*]Equal contribution. Order randomly determined.

method adds a progressive noise to the tuning procedure to obtain a distribution of effective prompts, while also maintaining the fluency of the prompts through a perplexity constraint. As shown in Figure 1, compared to the baseline gibberish prompts produced by AutoPrompt, FLUENTPROMPT generates prompts that are more fluent (i.e., lower perplexity) and perform competitively. The resulting fluent prompts not only facilitate our further analysis, but can also lead to better trust and engagement from both researchers and end users.

After obtaining a broad set of effective and human-readable prompts, we analyze the factors that contribute to the effectiveness of prompts. Specifically, we show that effective prompts are both (1) **topically related to the task domain** and (2) **more *calibrated* to the prior probability of output labels.** Specifically, calibration measures how balanced the LM's prior probability of output labels (i.e., in the absence of a specific example) is.

Based on our findings, we propose a novel method UNSUPERVISED FLUENTPROMPT, for automatically searching for effective prompts using only unlabeled data. UNSUPERVISED FLUENTPROMPT optimizes the prompts for both better calibration and better domain relevance. Our experimental results show that UNSUPERVISED FLUENTPROMPT outperforms strong zero-shot baseline (Holtzman et al., 2021) by 7.0% in accuracy. We summarize our contributions as follows:

- We introduce FLUENTPROMPT, a human-readable prompt tuning method that can generate a broad set of *effective* and *fluent* prompts (§3). This method not only serves as the foundation for our analysis, but also helps bridge the gap between manual prompt engineering and gradient-based prompt tuning.

- We analyze the factors that contribute to the effectiveness of prompts and show that topic relatedness and calibration of the prompts are key to their success (§4).

- Inspired by our findings, we introduce a new method for discovering effective prompts without the need for labeled data (§5).

## 2   Related Work

### 2.1   Prompt Tuning

**Continuous Prompt**   Continuous prompts are continuous vectors inserted to the task input for a prompted language model (Qin and Eisner, 2021; Ding et al., 2021; Lester et al., 2021; Liu et al., 2021). Such continuous prompts are typically tuned by gradient-based methods, which are guided by the task training examples with labels. While these prompts usually improve the model performance, their continuous nature makes them difficult for humans to understand or interpret (Khashabi et al., 2021; Hambardzumyan et al., 2021).

**Discrete Prompt**   Discrete prompts are composed of discrete tokens from natural language vocabulary. Such prompts can be either written by human or searched automatically. Human-written prompts (Kojima et al., 2022; Wang et al., 2022; Sanh et al., 2021; Su et al., 2022) typically consist of meaningful texts such as task descriptions (Schick and Schütze, 2021) or instructions (e.g., "let's think step by step", Kojima et al. 2022), which are not only human readable but also align with our understanding of tasks. In-context demonstration examples can also be considered as human-written prompts (Brown et al., 2020; Liu et al., 2022) but are not the focus of this work.

Prior work has also focused on searching discrete prompts automatically. One method is gradient-based similar to the continuous prompt setup but with projections to a discrete vocabulary (Shin et al., 2020). The drawback of this method is that the resulting prompts are usually disfluent and difficult to read. Other work searching for discrete prompts include edit-based enumeration (Prasad et al., 2022), reinforcement learning (Deng et al., 2022), and large language model continuation and filtering (Zhou et al., 2022). The goal for these prompt tuning methods is mainly to achieve competitive task performance without modifying language model parameters.

The purpose of our work is to analyze what aspects of the tuned natural language prompts make them effective for zero-shot inference of language models. To facilitate such analysis, we need prompt readability as in human-written prompts and also a large search space as in gradient-based discrete prompt tuning. FLUENTPROMPT bridges the gap and provides a distribution of *effective and human-readable* prompts.

### 2.2   Analyses of Prompts

A growing body of literature tries to understand the mechanisms behind prompts via various per-

spectives. For example, prompts in the form of in-context examples are analyzed under perturbations w.r.t. order, label, editing, etc. (Lu et al., 2022; Min et al., 2022; Chen et al., 2022). Human-written instructions (Mishra et al., 2021) have also been studied and show weak sensitivity to semantic-changing perturbations (Webson and Pavlick, 2021). Gonen et al. (2022) use paraphrasing and back-translation on a set of human-written prompts and find on these natural prompts there is a correlation between lower perplexity and better resulting performance.

Our work focuses on natural language prompts derived from gradient-based prompt tuning. Khashabi et al. (2021) tune continuous prompts and show that effective continuous prompts may transfer poorly to their nearest discrete prompts. In contrast, we perform prompt tuning in the discrete space directly with FLUENTPROMPT, demonstrating the feasibility of searching for readable prompts using gradient-based method. This approach gives us a more faithful understanding of the factors that contribute to the effectiveness of natural language prompts.

## 3 FLUENTPROMPT

We introduce FLUENTPROMPT, a prompt tuning method that generates a group of highly effective and human-readable prompts. Our approach utilizes Langevin dynamics to incorporate fluency constraints into the prompt tuning process, making it a novel application of controllable text generation and constrained sampling within the field of discrete prompt tuning. With human-readable prompts, we aim to explore the relationship between the features of the prompts and their performance.

### 3.1 Background: continuous prompt tuning

Given an input example $x$ with an output label $y \in Y$, we can *prompt* an autoregressive language model with parameters $\theta$ as follows. We reformulate the task as a language modeling problem by inserting a task-specific template $t$ to $x$, and defining a verbalizer $v$ mapping from a label $y$ to a label word (i.e, a token in the LM's vocabulary that semantically implies the label). For example, to determine the sentiment of "I like the movie", we can pass "I like the movie. It was [MASK]" to the LM and inspect the probability of "good" as a label word.[1] Specifically, the probability of a label $y$ given an input $x$ and template $t$ is estimated by:

$$p_\theta(v(y) \mid x, t) = \frac{\exp \text{logit}_\theta(v(y) \mid x, t)}{\sum_{y'} \exp \text{logit}_\theta(v(y') \mid x, t)}$$
(1)

Lester et al. (2021) add a sequence of $M$ soft embeddings $\tilde{e}_{0:M}$ (simplified as $\tilde{e}$; $0{:}M$ refers to the positional subscript for the sequence from position 0 to $M-1$) in front of the input. Therefore, the probability of the label is computed by $p_\theta(v(y) \mid \tilde{e}, x, t)$, where $\tilde{e}$ is embeddings that bypass the word embedding layer of the LM $\theta$ and is learned based on a set of training data. These learned embeddings are sometimes referred to as soft prompts, and the learning of such prompts as soft prompt tuning. For example, if stochastic gradient descent (SGD) is used as an optimizer, the soft prompt $\tilde{e}$ is updated as

$$\tilde{e}^i = \tilde{e}^{i-1} - \eta \nabla_{\tilde{e}}(-\log p_\theta(v(y) \mid \tilde{e}^{i-1}, x, t))$$
(2)

where $i$ is the timestep superscript, referring to $i$-th optimization step, and $\eta$ is the learning rate.

### 3.2 Method

#### 3.2.1 Discrete prompt tuning with Langevin dynamics

There are two challenges for the above soft prompt tuning. First, the resulting embeddings cannot be mapped to the natural language vocabulary. Khashabi et al. (2021) show that naively mapping an effective soft prompt to its nearest tokens significantly drops the performance. Second, we only obtain a single embedding instead of a range of embeddings with varying levels of performance. This makes it difficult to analyze the characteristics of the prompts and compare their effectiveness in specific tasks.

Following Kumar et al. (2022), we use Langevin dynamics to sample discrete prompts that lead to a better performing model in the task. Overall, the method is similar to SGD but adds a progressive Gaussian noise to the embeddings, with the scale decreasing over time. Additionally, at each optimization step, the updated embedding is projected to the nearest embedding in the LM vocabulary.

$$\tilde{e}^i = \text{Proj}_{\mathbf{E}}[\tilde{e}^{i-1} - \eta \nabla_{\tilde{e}} \mathcal{E}(\tilde{e}^{i-1}) + \sqrt{2\eta\beta_i} z]$$
(3)

---

[1] Table 7 shows the exact templates and verbalizers used throughout this work.

where:

- $\mathcal{E}$ is an energy function (lower is better), $\mathcal{E}(\tilde{\boldsymbol{e}}^{i-1}) = -\log p_\theta(v(y) \mid \tilde{\boldsymbol{e}}^{i-1}, \boldsymbol{x}, \boldsymbol{t})$.

- $\boldsymbol{z}$ is a Gaussian noise, $\boldsymbol{z} \sim \mathcal{N}(0, I_{|\tilde{e}|})$.

- $\beta$ is the variance of the noise following a geometric progression, $\beta_{\text{start}} > \beta_i > \beta_{\text{end}} \to 0$.

- $\mathbf{E}$ is the embedding table (layer) of the LM $\theta$, one embedding for each token in the vocabulary.

- $\text{Proj}_{\mathbf{E}}$ is a projection operation finding a nearest neighbor for each soft embedding in the LM's vocabulary, $\text{Proj}_{\mathbf{E}}(\tilde{e}) = \text{argmin}_{e_v \in \mathbf{E}}(\|e_v - \tilde{e}\|_2)$.

Without the progressive noise in Langevin dynamics, our prompt search procedure is gradient-based and shares a similar intuition with Auto-Prompt (Shin et al., 2020). Both methods use the gradient of the loss w.r.t. the embeddings, though AutoPrompt applies greedy substitution whereas we use projected gradient descent, aligning with soft prompt tuning and enabling the subsequent prompt sampling. AutoPrompt also incorporates verbalizer word selection, which is not a focus of the analysis in this work. We use our gradient-based, discrete prompt tuning method without Langevin dynamics as a baseline, referred to as $\text{AutoPrompt}_{\text{SGD}}$.

### 3.2.2 Fluency constraint

Sampling from projected Langevin dynamics ensures that the tuned prompt contains natural language tokens. However, with no extra constraints, they can form a disfluent sentence.

We explicitly incorporate a fluency objective to the Langevin energy function. This objective resembles the regular perplexity loss, but the labels (next token in the prompt) are not ground-truth. Instead, we measure an embedding-based sequence probability according to Kumar et al. (2022). For simplicity, below we drop the timestep superscript on the prompt embeddings and only keep the positional subscript.

The first step is to obtain the probability of generating the embedding at position $m$ (i.e., $\tilde{e}_m$) based on the previous $m-1$ embeddings (i.e., $\tilde{\boldsymbol{e}}_{0:m}$). We extract the last hidden state from the LM (i.e., output embedding) at position $m-1$: $h_{\theta,m-1} = h_\theta(\tilde{\boldsymbol{e}}_{0:m})$. Then the probability is:

$$p_\theta(\tilde{e}_m \mid \tilde{\boldsymbol{e}}_{0:m}) = \frac{\exp(h_{\theta,m-1} \cdot \tilde{e}_m)}{\sum_{e_v \in \mathbf{E}} \exp(h_{\theta,m-1} \cdot e_v)} \quad (4)$$

where we equivalently compute the logits for each embedding's corresponded vocabulary and take the softmax.[2] Subsequently, the sequence probability is $p_\theta(\tilde{\boldsymbol{e}}_{0:M}) = \prod_{m=1}^{M-1} p_\theta(\tilde{e}_m \mid \tilde{\boldsymbol{e}}_{0:m})$.

We define a prompt fluency loss as the negative log-likelihood of the prompt embeddings, $-\log p_\theta(\tilde{\boldsymbol{e}}_{0:M})$. Along with the task labeling loss (§3.2.1), we modify our energy function as:

$$\mathcal{E}(\tilde{\boldsymbol{e}}_{0:M}) = -\lambda_{\text{task}} \log p_\theta(v(y) \mid \tilde{\boldsymbol{e}}_{0:M}, \boldsymbol{x}, \boldsymbol{t})$$
$$- \lambda_{\text{fluency}} \log p_\theta(\tilde{\boldsymbol{e}}_{0:M}) \quad (5)$$

where $\lambda_{\text{task}} + \lambda_{\text{fluency}} = 1$. Through the whole FLUENTPROMPT tuning procedure, the language model parameters $\theta$ are fixed while the embeddings $\tilde{e}_{0:M}$ are tuned.

### 3.3 Experimental Setup

**Target tasks** We evaluate performance on two sentiment analysis tasks: Amazon Polarity (McAuley and Leskovec, 2013) and SST-2 (Socher et al., 2013), and one topic classification task: AGNEWS (Zhang et al., 2015). These tasks were selected since vanilla soft prompt tuning (Lester et al., 2021) substantially improves model performance. In contrast, tasks like RTE (Dagan et al., 2005) are more difficult; soft prompt tuning did not yield a significant improvement (57.4% accuracy from prompt tuning compared with 52.1% from random guess) in our pilot study, and we therefore did not pursue further analysis using FLUENTPROMPT. The verbalizer words and templates used for each task are listed in Table 7.

**Model** We optimize prompts for GPT-2 large (774M parameters, Radford et al. 2019) using FLUENTPROMPT. We use a batch size of 16 and train for 5,000 steps with an AdamW optimizer (Loshchilov and Hutter, 2018). We select the best prompt based on the validation performance. For our method FLUENTPROMPT, we use a step size $\eta \in \{0.3, 1.0, 3.0, 10.0\}$, $\beta_{\text{start}} = 1.0$, $\beta_{\text{end}} = 0.0001$, $\lambda_{\text{fluency}} \in \{0.003, 0.01, 0.03, 0.1, 0.3\}$. We search for both 5-token prompts ($M = 5$) and 10-token prompts ($M = 10$) and use 10 random seeds

---

[2]This is equivalently computing the logits since $e_v$ and the projected $\tilde{e}_m$ from the last optimization step are both in the embedding table.

| Prompt | Acc. | PPL |
|---|---|---|
| **SST-2** | | |
| **Empty Prompt** | 66.5 | - |
| **AutoPrompt$_{\text{SGD}}$** | | |
| Compl disgustingÃÂÃÂ Rated jer | 87.6 | $> 10^6$ |
| **FLUENTPROMPT** | | |
| Kubrick, "The Shining | 87.5 | 13.1 |
| Paramount, "The Shining | 86.8 | 12.2 |
| Kubrick\'s "The Man | 86.3 | 9.3 |
| disappointing.\n\n" | 84.4 | 4.1 |
| **AMAZON** | | |
| **Empty Prompt** | 75.8 | - |
| **AutoPrompt$_{\text{SGD}}$** | | |
| Reviewed experien audition lashesrible | 82.2 | $> 10^6$ |
| **FLUENTPROMPT** | | |
| scathing.\n\n" | 83.1 | 5.1 |
| upset.\n\n" | 82.6 | 3.67 |
| cigars: \n\n | 82.4 | 20.9 |
| mascara\n\n\n | 82.2 | 47.1 |
| **AGNEWS** | | |
| **Empty Prompt** | 49.7 | - |
| **AutoPrompt$_{\text{SGD}}$** | | |
| EStreamFramenetflixnetflixobookgenre | 69.3 | $> 10^5$ |
| **FLUENTPROMPT** | | |
| netflix/genre/netflix | 71.1 | 281.0 |
| netflix AnimeMoviegenre\n | 70.1 | 1925.0 |
| Synopsis\n\nThe story is | 69.2 | 9.6 |
| pmwiki.php/main/Superhero | 65.0 | 2.4 |

Table 1: Accuracy (*Acc.*) and Perplexity (*PPL*) of prompts. Both FLUENTPROMPT and AutoPrompt$_{\text{SGD}}$ use $M$=5 tunable tokens. FLUENTPROMPT shows comparable performance to the AutoPrompt$_{\text{SGD}}$ but with significantly lower perplexity. Prompts discovered by FLUENTPROMPT show domain relevance and potential calibration for model outputs.

for each hyperparameter setup. Additionally, we perform experiments with $\beta_{\text{start}} = \beta_{\text{end}} = 0$ (i.e, no progressive noise) and $\lambda_{\text{fluency}} = 0$ (i.e, no fluency constraint) as ablations to FLUENTPROMPT purposed for analysis.

### 3.4 Results

Table 1 shows example prompts found by AutoPrompt$_{\text{SGD}}$ and FLUENTPROMPT, along with their associating accuracy and perplexity. We additional show the accuracy of an *empty* prompt (i.e., $\tilde{e}$ is null). We see that FLUENTPROMPT performs comparably to AutoPrompt$_{\text{SGD}}$ and significantly better than the empty prompt. In terms of readability, FLUENTPROMPT generates more fluent prompts than AutoPrompt$_{\text{SGD}}$.

In Table 2, we quantitatively compare AutoPrompt$_{\text{SGD}}$ and FLUENTPROMPT. For each task, we use each method to generate 40 prompts with a length $M = 10$, under 4 step sizes

$\eta \in \{0.3, 1.0, 3.0, 10.0\}$ and 10 random seeds. AutoPrompt$_{\text{SGD}}$ does not have a perplexity constraint over its prompt tuning process ($\lambda_{\text{fluency}} = 0$). For FLUENTPROMPT, we apply an optimal perplexity constraint at $\lambda_{\text{fluency}} = 0.003, 0.003, 0.01$ for SST-2, Amazon, and AGNEWS, respectively. We observe that on Amazon, FLUENTPROMPT achieves both a better average prompt accuracy and a better maximum accuracy. On SST-2 and AGNEWS, FLUENTPROMPT also achieves better average accuracy, while having a nearly as high maximum accuracy as AutoPrompt$_{\text{SGD}}$. For all three tasks, FLUENTPROMPT leads to prompts with a significantly lower perplexity ($p < 0.0001$ in $t$-tests). Without sacrificing performance, prompts with lower perplexity are preferred for their potentially better readability for downstream analyses.

## 4 What Makes Good Prompts?

In this section, we analyze common attributes of the effective tuned prompts. Specifically, we study the 10-token prompts found by FLUENTPROMPT on SST-2, Amazon and AGNEWS.

### 4.1 Effective prompts calibrate the output distribution over label words

Language models are known to be biased towards label words that are common in its pretraining distribution (Holtzman et al., 2021; Zhao et al., 2021). In this section, we aim to investigate whether effective prompts found by prompt tuning implicitly adjust for the bias (calibration). To measure this bias, we follow Holtzman et al. (2021) to use task-specific domain string $\boldsymbol{d}$ as the test input and compute the entropy of the labels. Table 3 lists the task-specific domain $\boldsymbol{d}$ for each dataset. As the task-specific domain strings do not imply any label information (i.e., label-neutral), we expect the output of the language model to be uniform over the label words when only conditioned on the domain string. The entropy of the label words is computed as follows:

$$H(y) = \mathbb{E}_{y \in Y}[-\log p(y)] =$$
$$- \sum_{y \in Y} p_\theta(v(y) \mid \tilde{\boldsymbol{e}}, \boldsymbol{d}, \boldsymbol{t}) \log p_\theta(v(y) \mid \tilde{\boldsymbol{e}}, \boldsymbol{d}, \boldsymbol{t})$$
$$(6)$$

Higher entropy of the label word prediction implies a more balanced (calibrated) label words distribu-

| | SST-2 | | | Amazon | | | AGNEWS | | |
|---|---|---|---|---|---|---|---|---|---|
| | Mean Acc | Max Acc | log PPL | Mean Acc | Max Acc | log PPL | Mean Acc | Max Acc | log PPL |
| AutoPrompt$_{\text{SGD}}$ | 84.99 | **90.48** | 13.89 | 83.36 | 86.95 | 13.33 | 69.74 | **80.50** | 15.68 |
| FLUENTPROMPT | **87.54** | 90.14 | **9.27** | **85.20** | **88.20** | **10.20** | **73.34** | 79.50 | **10.93** |

Table 2: Prompt effectiveness and perplexity of AutoPrompt$_{\text{SGD}}$ and FLUENTPROMPT. Each model derives 40 prompts with a length of 10. AutoPrompt$_{\text{SGD}}$ does not have the progressive noise $z$ and the perplexity constraint ($\lambda_{\text{fluency}} = 0$). FLUENTPROMPT applies the perplexity constraint with $\lambda_{\text{fluency}} = 0.003, 0.003, 0.01$ for SST-2, Amazon, and AGNEWS, respectively. **The prompts found by FLUENTPROMPT are overall more *effective* and have a significantly lower perplexity, indicating *better readability*.**

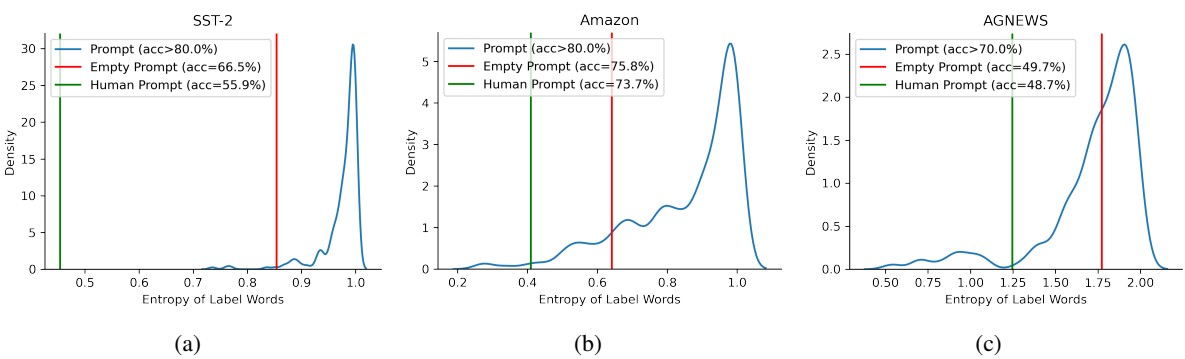

Figure 2: Frequency of prompts (y-axis) at different entropy levels (x-axis). We compare effective prompts with the empty and human-written prompt.

| Task | Domain String $d$ |
|---|---|
| SST-2 | This is a movie review |
| Amazon | This is an Amazon product review |
| AGNEWS | This is a news |

Table 3: Tasks and their task-specific domain strings. The task-specific domain strings do not imply any label information.

tion. When the label word probabilities are uniform, the entropy reaches its maximum at $\log(|Y|)$.

As listed in Table 1, some effective prompts found by FLUENTPROMPT for sentiment analysis contain negative sentiment words (e.g., "disappointing" and "complained" in prompts for SST-2 ), which may implicitly reduce the probabilty of positive labels and calibrate the label word distribution. To validate this hypothesis, we filter a set of effective prompts by FLUENTPROMPT and compute the entropy of the label predictions conditioned on the concatenation of the prompt and the task-specific domain string. Figure 2 shows the density plot comparing the label word entropy of effective prompts, with empty and human-written prompts taken from Bach et al. (2022). We observe that the entropy of effective prompts has a higher mode than the entropy of empty and human-written prompts with

lower accuracy.

To further explore the relation between the task performance and calibration, we compute correlation between the task accuracy and the label word entropy of all prompts obtained by FLUENTPROMPT and report Spearman's rank correlation. From Figure 3, we observe that the label word entropy exhibits significant positive correlations with the task accuracy (all $p < 0.0001$). The Spearman's coefficients are +0.61, +0.75 and +0.43 for SST-2, Amazon and AGNEWS, respectively.

## 4.2 Effective prompts are topically related to the task domain

**Qualitative Analysis** As shown in Table 1, most of the effective prompts obtained by FLUENTPROMPT contain domain-related words. For example, the prompt *Kubrick, "The Shining* in SST-2 contains a movie director name and a movie title, relevant to the domain of movie reviews. Similarly, the prompts *mascara\n\n* and *cigars\n\n* found for the Amazon dataset contain product names relevant to the domain of product reviews. Additionally, AGNEWS is a news topic classification task. Some of the effective prompts in AGNEWS contain topic classification-related words such as "genre", while others contain URLs that

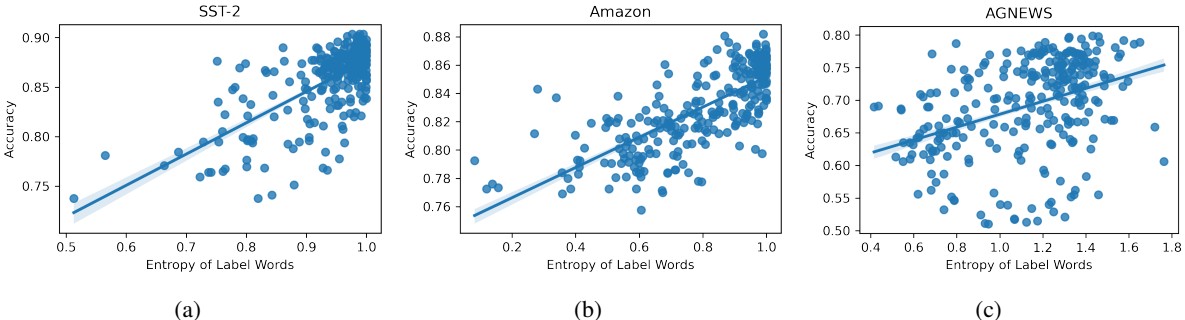

Figure 3: Correlation between task performance and label word entropy. Spearman rank correlation coefficients for SST-2, Amazon and AGNEWS are +0.42, +0.76 and +0.57. All p-values are smaller than 0.0001.

link to websites such as netflix[3] and pmwiki.[4] The target pages of these URLs also contain topic classification-related information, such as the prompt *pmwiki/pmwiki.php/Main/Superhero* which links to a wiki page containing the following information: "*Genre: Action Adventure Comedy Commercials*".

**Quantitative Analysis**  Based on our qualitative analysis, we hypothesize that effective prompts are topically related to the task domain. To validate this hypothesis, we compare domain word frequency in effective prompts and random sentences. First, we select a set of domain words for each task (see Table 4), which consist of the task label words (e.g., "positive" and "negative" for SST-2) and common words in the task domain (e.g., "movie" and "film" for the movie domain of SST-2). Since our prompts are very short (10 tokens), we augment each prompt with its continuation generated by GPT-3 (Brown et al., 2020), based on the assumption that the continuation by the large LM follows the same domain as the prompt. For each prompt, we sample 5 distinct continuations from GPT-3 using nucleus sampling $p = 0.95$ at a length of 100 tokens. We compare the top 10 effective prompts with 10 random sentences from PILE (Gao et al., 2020) augmented by the same continuations. We then count the domain words in the concatenation of the prompt and its continuation.

Table 5 lists the average accuracy and the number of domain words in the effective and the random sentences with their continuations. The accuracy of effective prompts is higher than that of random sentences on all three datasets. Moreover, the domain words frequency of effective prompts is

| Task | Domain Words |
|------|--------------|
| SST-2 | movie, film, cinima, director, positive, negative |
| Amazon | book, amazon, product, furniture, positive, negative |
| AGNEWS | topic, category, politics, sports, business, technology |

Table 4: Domain words for each task.

| | SST-2 | | Amazon | | AGNEWS | |
|---|---|---|---|---|---|---|
| | Acc. | Freq. | Acc. | Freq. | Acc. | Freq. |
| Effective | 89.4 | 23.4 | 86.5 | 5.8 | 77.6 | 3.7 |
| Random | 67.2 | 1.3 | 74.2 | 2.2 | 49.3 | 0.8 |

Table 5: Average domain words frequency (Freq.) and average accuracy (Acc.) for effective prompts and random sentences. **Effective prompts and their continuation contain substantially more domain words than random sentences and their continuation.** The p-values from the paired t-test for SST-2, Amazon, and AGNEWS were 0.004, 0.003, and 0.0002, respectively.

significantly higher than that of random sentences with p-values of 0.004, 0.003, and 0.0002 for SST-2, Amazon, and AGNEWS, respectively. Both our qualitative and quantitative analysis provide strong evidence that effective prompts obtained by our prompt tuning are topically related to the task's domain.

## 5  UNSUPERVISED FLUENTPROMPT

Our analysis in Section 3 shows that effective prompts exhibit calibration and have high domain relevance to the task. Since these two features are both highly indicative and do not require ground-truth labels for computation, we propose UNSU-PERVISED FLUENTPROMPT, a method for automatically identifying effective prompts **without labeled data**. The key idea is to optimize the prompts

[3]www.netflix.com
[4]www.pmwiki.org

for improved calibration and domain relevance. In the following sections, we will detail the methodology of UNSUPERVISED FLUENTPROMPT (§5.1), describe the experimental setup (§5.2), and present the results (§5.3).

## 5.1 Method

**Calibration loss** In Section 4.1, we find a strong positive correlation between the degree of calibration and performance of the prompts. We therefore explicitly optimize the prompt towards greater calibration (i.e., maximizing the entropy of label words). Ideally, we need a large set of label-neutral domain strings to prevent the model from learning noises in the procedure. Since these domain strings are not always easy to obtain, we use the training inputs of the task (without ground-truth labels), and expect that the aggregation of them should be label-neutral. Therefore, we define a calibration loss based on the entropy of the label words distribution:

$$\mathcal{L}_{\text{entropy}}(\tilde{\boldsymbol{e}}) = \mathbb{E}_{y \in Y}[\log \mathbb{E}_{\boldsymbol{x} \in X} p_\theta(v(y) \mid \tilde{\boldsymbol{e}}, \boldsymbol{x}, \boldsymbol{t})]$$

Intuitively, the calibration loss encourages the prompt to help the model generate more balanced predictions at a dataset (macro) level rather than instance (micro) level.

**Domain relevance loss** In Section 4.2, we find that effective prompts are overall more related to the task domain. To explicitly make the prompt relevant to the domain, we extend the existing fluency (perplexity) loss from Section 3.2.2, modeling the perplexity of both the prepending prompt and the input example:

$$\mathcal{L}_{\text{domain}}(\tilde{\boldsymbol{e}}) = - \log p_\theta(\tilde{\boldsymbol{e}}_{0:M}) \tag{7}$$

$$- \sum_i \log p_\theta(x_i \mid \tilde{\boldsymbol{e}}, \boldsymbol{x}_{<i}) \tag{8}$$

$$- \sum_j \log p_\theta(t_j \mid \tilde{\boldsymbol{e}}, \boldsymbol{x}, \boldsymbol{t}_{<j}) \tag{9}$$

Intuitively, $\log p_\theta(\boldsymbol{x} \mid \tilde{\boldsymbol{e}}) - \log p_\theta(\boldsymbol{x})$ would measure the pointwise mutual information between the task data $\boldsymbol{x}$ and the tuned prompt $\tilde{\boldsymbol{e}}$, with the part $\log p_\theta(\boldsymbol{x})$ not involved in the prompt optimization.

Overall, our unsupervised energy function $\mathcal{E}$ is updated to:

$$\mathcal{E}(\tilde{\boldsymbol{e}}_{0:M}) = - \lambda_{\text{calibration}} \mathcal{L}_{\text{entropy}}(\tilde{\boldsymbol{e}}) \tag{10}$$

$$- \lambda_{\text{domain}} \mathcal{L}_{\text{domain}}(\tilde{\boldsymbol{e}}) \tag{11}$$

where $\lambda_{\text{calibration}} + \lambda_{\text{domain}} = 1$.

|  | SST-2 | Amazon | AGNEWS |
|---|---|---|---|
| **Unsupervised Method** | | | |
| Emtpy | 66.5 | 75.8 | 49.7 |
| PMI$_{\text{DC}}$ | 85.6 | 76.2 | 64.1 |
| UNSUP. F.P. | **88.2** | **85.3** | **68.0** |

Table 6: Accuracy of different unsupervised prompting methods on the three datasets. UNSUP. F.P. refers to our UNSUPERVISED FLUENTPROMPT.

## 5.2 Experimental Setup

Inheriting the notations of FLUENTPROMPT, we consider the following hyperparameters: $\eta \in \{1.0, 3.0\}$, $\beta_{\text{start}} = 1.0$, $\beta_{\text{end}} = 0.0001$, $\lambda_{\text{domain}} \in \{0, 0.0003, 0.001, 0.003, 0.01, 0.05, 0.2, 0.5\}$, $M = 10$. We use five random seeds for each setup and report the average performance.

## 5.3 Results

In Table 6, we compare the performance of our proposed method, UNSUPERVISED FLUENTPROMPT, with the empty prompt and the strong unsupervised baseline PMI calibration PMI$_{\text{DC}}$ (Holtzman et al., 2021) on three datasets. Our results show that UNSUPERVISED FLUENTPROMPT consistently outperforms PMI$_{\text{DC}}$ with an average improvement of 7.0% across the datasets. This demonstrates that the incorporated calibration and domain information are helpful to finding effective prompts.

## 6 Conclusion

In this paper, we investigate the factors that contribute to the effectiveness of prompts. To facilitate this study, we develop a human-readable prompt tuning method FLUENTPROMPT and apply it to the GPT-2 large model to generate effective and readable prompts. Our analysis reveals that effective prompts are topically related to the task domain and calibrate the prior probability of label words.

Although the prompts generated by FLUENTPROMPT are effective and readable, they still carry limited semantic meanings. For instance, we did not find any prompts directly indicating the task definition or instructions. One potential reason is that the GPT-2 large model is not instruction-tuned. Future work can apply FLUENTPROMPT to an instruction-tuned model to see if instruction-like prompts can be discovered.

## Limitations

The FLUENTPROMPT approach is a versatile method for optimizing human-readable prompts in both classification and generation tasks. However, our investigations into calibration are specific to classification tasks. It would be intriguing to explore the characteristics of effective prompts in generation tasks in future studies. It is worth noting that FLUENTPROMPT employs Langevin dynamics to incorporate perplexity constraints during training, making it directly applicable to autoregressive models and not to masked language models or encoder-decoder models.

Due to resource limitations, we applied FLUENTPROMPT to GPT-2 large. Our current GPU was not sufficiently efficient to handle larger models within our budget. It is important to note that our focus was not solely on performance, but rather on analyzing the properties of effective prompts. In the future, it would be valuable to extend our method to different-sized language models and explore alternative constrained sampling techniques to identify fluent and effective prompts for various types of language models.

## Acknowledgements

We gratefully acknowledge support from NSF CAREER Grant No. IIS2142739. This material is funded in part by the DARPA Grant under Contract No. HR001120C0124. This research is supported in part by the Office of the Director of National Intelligence (ODNI), Intelligence Advanced Research Projects Activity (IARPA), via the HIATUS Program contract #2022-22072200004. The views and conclusions contained herein are those of the authors and should not be interpreted as necessarily representing the official policies, either expressed or implied, of ODNI, IARPA, or the U.S. Government. The U.S. Government is authorized to reproduce and distribute reprints for governmental purposes notwithstanding any copyright annotation therein.

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

## A  Verbalizer and templates

Table 7 shows an example input, template and the verbalizer used for each task.

| Task | Template | Label words through verbalizer |
|---|---|---|
| SST-2 | Illuminating if overly talky documentary. It was | positive, negative |
| Amazon | Terrible service. It was | positive, negative |
| AGNEWS | Economic growth in Japan slows down as the country experiences. It is about | politics, sports, business, technology |

Table 7: The template, example (colored black) and verbalizer used for each dataset.