# OpenReview forum: "Toward Human Readable Prompt Tuning: Kubrick’s The Shining is a good movie, and a good prompt too?"
_EMNLP/2023/Conference — EMNLP 2023 Findings_

### Official Review · Reviewer_ceBW · 2023-08-04

**Soundness:** 4

**Excitement:**

4: Strong: This paper deepens the understanding of some phenomenon or lowers the barriers to an existing research direction.

**Paper Topic And Main Contributions:**

This paper describes a new method for automatic prompt-tuning which produces more interpretable prompts than existing methods. Specifically, the authors describe a gradient-based method for finding the best prompt for a task, but their method includes a fluency objective that constrains the model to prefer token sequences with low perplexity. The method thus produces prompts that acheive similar performance to those found using unconstrained automatic prompt tuning methods, but has the added benefit that the discovered prompts are (somewhat) interpretable to humans. The authors then use the method to perform some analysis on prompts, and which aspects of prompts produce good results. They reveal some interesting and intuitive insights. E.g., they show that topic keywords in the prompt help to guide the model, and that words are often included in the prompt which prime the model to have the right distribution over labels (e.g., negative sentiment words to counteract what might otherwise be a bias toward affirmative answers).

This is a nice paper overall. The method is nice and I really appreciate the analysis the authors perform. There is a lot of material and it holds together nicely. I have no major concerns, just a few questions about clarity.

**Questions For The Authors:**

* Can you include a short description of what Langevin Dynamics is? I had to Google it. You shouldn't assume your reader has even a vague knowledge of what that is. You don't need to go into details (I was perfectly able to follow the paper without knowing anything about Langevin, so you could honestly cut this connection entirely). But if you are going to bring it up, you should describe what it is.
* Sorry if this is a stupid question--but I struggled to follow why you define your loss over the embeddings (probability of the embeddings) rather than just over the tokens themselves, as is more standard? There must be some reason for this (perhaps related to the langevin dynamics?) but it was not clear to me. I think providing some better intuition for why your loss is defined the way it is would be important.
* I was confused initially in section 3.2 because I thought you were describing an unsupervised model (as this was emphasized in your intro). But actually 3.2 describes a supervised model, and its only at the end of the paper that you back off to the unsupervised one. Organizationally, the presentation could be improved to prevent this confusion. It would help to give more details about the data you are using and the loss in section 3.2, and make explicit that you will describe the unsupervised version later.
* The unsupervised results are (perhaps too!) good. IIUC that method actually performs about the same as the supervised one. It makes me suspicious that the method is actually behaving like autoprompt, rather than like fluentprompt. Can we see some example prompts discovered by the unsupervised method, and verify that they are indeed more interpretable? Or is it finding pathological prompts that are just high-performing?


**Reasons To Accept:**

* Interesting methodological contribution for automatic prompt tuning
* Nice analysis on prompts in general which could generate some compelling follow up work
* Lots of experiments, its a hefty paper


**Reasons To Reject:**

* Nothing really. I found the description of the method hard to follow and some clarifications would be nice. See Questions section below.

**Reproducibility:**

3: Could reproduce the results with some difficulty. The settings of parameters are underspecified or subjectively determined; the training/evaluation data are not widely available.

**Reviewer Confidence:**

3: Pretty sure, but there's a chance I missed something. Although I have a good feel for this area in general, I did not carefully check the paper's details, e.g., the math, experimental design, or novelty.

---

> ### Author Rebuttal · Authors · 2023-08-29
>
> Due to the number of reviews we got for this work, we have to prioritize some common questions raised in all reviews and make a general response to all reviewers. However, we will be super happy to discuss and follow up here if you think any of your crucial concerns are not addressed. Thank you so much! We additionally thank your suggestions on presentation improvements.

---

### Official Review · Reviewer_mZkP · 2023-08-05

**Typos Grammar Style And Presentation Improvements:** N/A
**Soundness:** 3

**Excitement:**

3: Ambivalent: It has merits (e.g., it reports state-of-the-art results, the idea is nice), but there are key weaknesses (e.g., it describes incremental work), and it can significantly benefit from another round of revision. However, I won't object to accepting it if my co-reviewers champion it.

**Missing References:**

N/A

**Paper Topic And Main Contributions:**

This manuscript concentrates on the prompt design within the prompt learning paradigm, aiming to enhance the interpretability and effectiveness of the prompt. The authors replace the stochastic gradient descent utilized in previous prompting optimization methods with Langevin dynamics and introduce fluency regulation during prompt training. They conduct experiments utilizing three datasets, namely Amazon Polarity, SST-2, and AGNEWS. The authors also analyze the shared characteristics of an effective prompt and examine the factors that contribute to prompt efficacy.

**Questions For The Authors:**

1.How are the domain words for each task obtained?
2.How many prompts are generated to obtain the analysis results shown in figure 2 and figure 3? How do you get various prompt for one task? By using different hyper-parameter?

**Reasons To Accept:**

1.This manuscript presents a new approach, FLUENTPROMPT, for tuning human-readable prompts in large language models. The method incorporates a fluency constraint and Langevin dynamics to generate effective and comprehensible prompts.
2.The manuscript offers a thorough analysis and exploration of the factors that contribute to prompt effectiveness, such as topical relevance to the task domain and calibration of the prior probability of output labels.
3.The manuscript proposes an unsupervised prompt tuning technique that optimizes prompts for better calibration and domain relevance using only unlabeled data. Empirical findings demonstrate that UNSUPERVISED FLUENT PROMPT outperforms robust zero-shot baselines in terms of accuracy.

**Reasons To Reject:**

1.The manuscript lacks sufficient motivation for the substitution of SGD with Langevin dynamics. It is unclear why introducing additional Gaussian noise would aid in this task.
2.The analysis of the common attributes of effective tuned prompts is somewhat flawed. Specifically, in quantitative analysis, GPT-3 is utilized to continue the prompt, and the word frequency is calculated based on the continuation. However, there is no assurance that the continuation follows the same distribution as the prompt. In reality, it follows the distribution of the GPT-3 pretraining corpus.
3.The ablation study of the domain loss and entropy loss in the proposed UNSUPERVISED FLUENTPROMPT is insufficient. Readers have no insight into which one contributes the most to the performance.
4.The authors only compare the performance of the proposed method with AutoPrompt. I believe that more baseline methods should be included to gain a better understanding of the effectiveness of the proposal. For instance, the following works also focus on discrete prompts:
a)RLPrompt: Optimizing Discrete Text Prompts with Reinforcement Learning
b)Black-box tuning for language-model-as-a-service
c)Grips: Gradient-free, edit-based instruction search for prompting large language models.

**Reproducibility:**

4: Could mostly reproduce the results, but there may be some variation because of sample variance or minor variations in their interpretation of the protocol or method.

**Reviewer Confidence:**

3: Pretty sure, but there's a chance I missed something. Although I have a good feel for this area in general, I did not carefully check the paper's details, e.g., the math, experimental design, or novelty.

---

> ### Author Rebuttal · Authors · 2023-08-29
>
> Due to the number of reviews we got for this work, we have to prioritize some common questions raised in all reviews and make a general response to all reviewers. However, we will be super happy to discuss and follow up here if you think any of your crucial concerns are not addressed. Thank you so much! We additionally thank your suggestions on presentation improvements.

---

### Official Review · Reviewer_xFvG · 2023-08-11

**Typos Grammar Style And Presentation Improvements:** 1. L028
**Soundness:** 4

**Excitement:**

4: Strong: This paper deepens the understanding of some phenomenon or lowers the barriers to an existing research direction.

**Paper Topic And Main Contributions:**

The authors address the problem of prompt tuning, proposing FluentPrompt to generate human-readable prompts in natural language. They experiment with three datasets concerning sentiment aisles and topic classification, demonstrating that their proposed method outperforms the previous solution and generates more readable prompts. Furthermore, the authors perform multiple studies about prompts' calibration and topic-relatedness.

**Questions For The Authors:**

1. Could the samples in Table 1 be cherry-picked? The authors should also provide examples of bad prompts for a better analysis.


**Reasons To Accept:**

1. The paper is well-written and easy to follow.
2. The analysis is throughout and well performed.
3. The work addresses an important and interesting topic for the NLP community.

**Reasons To Reject:**

1. The experimental setup regards only one model (GPT-2). More models should be tested to verify the proposed solution's performance and generalizability.

Minor points:

2. Although the authors mention in the "Limitation" section why they did not adopt larger models than GPT-2, they should also describe the hardware employed for the experimental phase.
3. The authors should provide statistics of the benchmarked datasets.

**Reproducibility:**

3: Could reproduce the results with some difficulty. The settings of parameters are underspecified or subjectively determined; the training/evaluation data are not widely available.

**Reviewer Confidence:**

3: Pretty sure, but there's a chance I missed something. Although I have a good feel for this area in general, I did not carefully check the paper's details, e.g., the math, experimental design, or novelty.

---

> ### Author Rebuttal · Authors · 2023-08-29
>
> Due to the number of reviews we got for this work, we have to prioritize some common questions raised in all reviews and make a general response to all reviewers. However, we will be super happy to discuss and follow up here if you think any of your crucial concerns are not addressed. Thank you so much! We additionally thank your suggestions on presentation improvements.

---

### Official Review · Reviewer_oTX2 · 2023-08-11

**Soundness:** 5

**Excitement:**

4: Strong: This paper deepens the understanding of some phenomenon or lowers the barriers to an existing research direction.

**Paper Topic And Main Contributions:**

This paper is about investigating the factors that make prompts effective in natural language processing tasks, particularly in classification problems. The paper proposes a human-readable prompt tuning method called FLUENTPROMPT, which incorporates a fluency constraint to find a distribution of effective and fluent prompts. The authors analyze the effectiveness of prompts and show that topic relatedness and calibration of the prompts are key to their success. Based on their findings, they also propose a method for generating prompts using only unlabeled data, which outperforms strong baselines by an average of 7.0% accuracy across three tasks. The main contributions of this paper are the proposed FLUENTPROMPT method for generating effective and fluent prompts, the analysis of the factors that contribute to the effectiveness of prompts, and the proposed method for generating prompts using only unlabeled data.


**Reasons To Accept:**

The strengths of this paper include the proposed FLUENTPROMPT method for generating effective and fluent prompts, the analysis of the factors that contribute to the effectiveness of prompts, and the proposed method for generating prompts using only unlabeled data. The paper also provides insights into the importance of topic relatedness and calibration of prompts for their success. If this paper were to be presented at the conference or accepted into Findings, the main benefits to the NLP community would be the availability of a new method for generating effective and fluent prompts, which could improve the performance of natural language processing tasks. The paper's analysis of the factors that contribute to the effectiveness of prompts could also inform future research in this area. Additionally, the proposed method for generating prompts using only unlabeled data could be particularly useful for low-resource languages or situations where labeled data is scarce.

**Reasons To Reject:**

One potential weakness of this paper could be the limited scope of the analysis, as the authors only investigate the effectiveness of prompts in classification problems. Additionally, the proposed FLUENTPROMPT method may not be applicable to all natural language processing tasks, and the proposed method for generating prompts using only unlabeled data may not be as effective in certain situations. The main risk of having this paper presented at the conference or accepted into Findings could be that the proposed methods may not be as effective as expected in real-world scenarios, or that the analysis may not be generalizable to other natural language processing tasks beyond classification problems.

**Reproducibility:**

3: Could reproduce the results with some difficulty. The settings of parameters are underspecified or subjectively determined; the training/evaluation data are not widely available.

**Reviewer Confidence:**

2: Willing to defend my evaluation, but it is fairly likely that I missed some details, didn't understand some central points, or can't be sure about the novelty of the work.

---

> ### Author Rebuttal · Authors · 2023-08-29
>
> Due to the number of reviews we got for this work, we have to prioritize some common questions raised in all reviews and make a general response to all reviewers. However, we will be super happy to discuss and follow up here if you think any of your crucial concerns are not addressed. Thank you so much! We additionally thank your suggestions on presentation improvements.

---

### Official Review · Reviewer_Uwpm · 2023-08-12

**Soundness:** 3

**Excitement:**

4: Strong: This paper deepens the understanding of some phenomenon or lowers the barriers to an existing research direction.

**Paper Topic And Main Contributions:**

This paper explores methods of automatic prompt generation using gradient-based prompt tuning. It focuses on discrete rather than continuous prompt tuning, with a focus on human readability as well as improved accuracy. Its main contributions are:

1. A prompt tuning method that generates human readable prompts
2. An unsupervised method of generating prompts that does not require labelled data
3. Analysis of the properties of effective automatically generated prompts. The authors observe that good prompts tend to be 1) related in topic to the task domain and 2) calibrate model output to adjust for label bias.



**Questions For The Authors:**

A. I am unsure if I have correctly understood Table 5, which displays accuracies for effective prompts that are higher than Table 2. If Table 5 shows effective prompts + GPT-3 continuations (resulting in a 100 token prompt), then does this imply that augmenting prompts with LLMs leads to improved performance over your FluentPrompt method?

**Reasons To Accept:**

Good insights: Section 4 (What makes good prompts?) offers greater understanding of the properties of good prompts (topic relevance + label word calibration). This can aid the NLP community in understanding why prompt tuning is effective.

Novel methods: the paper proposes two methods (supervised and unsupervised) for prompt tuning that outperform baselines. In addition to accuracy increase, the human-fluency achieved by these generated prompts can aid researchers in interpreting their prompts and lead to greater awareness of potential model/dataset issues (e.g. uncovering label word bias).



**Reasons To Reject:**

1.Proposed method is not entirely successful: as noted by the authors, despite achieving lower perplexity than baselines, the 'human-readable' prompts remain somewhat cryptic and limited in semantic meaning.

2.Lack of comprehensiveness: the FluentPrompt method proposed is only compared to Empty Prompt and AutoPrompt baselines. While the authors' focus was on discrete prompt tuning, I would have liked to have seen the performance of continuous soft prompt tuning on these tasks, so as to demonstrate whether there is a tradeoff between readability (achieved via discrete prompts) and performance.





**Reproducibility:**

4: Could mostly reproduce the results, but there may be some variation because of sample variance or minor variations in their interpretation of the protocol or method.

**Reviewer Confidence:**

3: Pretty sure, but there's a chance I missed something. Although I have a good feel for this area in general, I did not carefully check the paper's details, e.g., the math, experimental design, or novelty.

---

> ### Author Rebuttal · Authors · 2023-08-29
>
> Due to the number of reviews we got for this work, we have to prioritize some common questions raised in all reviews and make a general response to all reviewers. However, we will be super happy to discuss and follow up here if you think any of your crucial concerns are not addressed. Thank you so much! We additionally thank your suggestions on presentation improvements.

---

### Official Review · Reviewer_RhgP · 2023-08-12

**Soundness:** 2

**Excitement:**

3: Ambivalent: It has merits (e.g., it reports state-of-the-art results, the idea is nice), but there are key weaknesses (e.g., it describes incremental work), and it can significantly benefit from another round of revision. However, I won't object to accepting it if my co-reviewers champion it.

**Paper Topic And Main Contributions:**

This study presents a human readable prompt tuning approach for LLMs, aiming to generate prompts that are both effective and fluent.


**Reasons To Accept:**

- The author conducted experimental analysis on "What makes good prompts."

- An enhanced method is proposed for automatically identifying effective prompts without labels.

**Reasons To Reject:**

- The implementation is based on GPT-2, which is not currently the state-of-the-art method, thereby diminishing the contribution of the work.
- This work lacks ablation analysis to analysis sereval crucial hyper-parameters, such as the lambda.
- The results of UNSUPERVISED FLUENTPROMPT are presented without substantial discussion.


**Reproducibility:**

2: Would be hard pressed to reproduce the results. The contribution depends on data that are simply not available outside the author's institution or consortium; not enough details are provided.

**Reviewer Confidence:**

2: Willing to defend my evaluation, but it is fairly likely that I missed some details, didn't understand some central points, or can't be sure about the novelty of the work.

**Typos Grammar Style And Presentation Improvements:**

 The formatting in the appendix appears to have some issues.

---

> ### Author Rebuttal · Authors · 2023-08-29
>
> Due to the number of reviews we got for this work, we have to prioritize some common questions raised in all reviews and make a general response to all reviewers. However, we will be super happy to discuss and follow up here if you think any of your crucial concerns are not addressed. Thank you so much! We additionally thank your suggestions on presentation improvements.

---

### Official Review · Reviewer_HPcZ · 2023-08-12

**Soundness:** 3

**Excitement:**

3: Ambivalent: It has merits (e.g., it reports state-of-the-art results, the idea is nice), but there are key weaknesses (e.g., it describes incremental work), and it can significantly benefit from another round of revision. However, I won't object to accepting it if my co-reviewers champion it.

**Missing References:**

Automatic Chain of Thought Prompting in Large Language Models

Large Language Models Are Human-Level Prompt Engineers


**Paper Topic And Main Contributions:**

This paper solves the problem about prompt interpretability, and introduce FLUENTPROMPTF, a human-readable prompt tuning method that can generate a broad set of effective and fluent prompts, bridging the gap between manual prompt engineering and gradient-based prompt tuning. Besides, they analyze the factors that contribute to the effectiveness of prompts and show that topic relatedness and calibration of the prompts are key to their success.

**Reasons To Accept:**

The approach is both straightforward and efficient, effectively addressing the issue of prompt interpretability which holds substantial practical significance.

The paper not only analyzes the elements contributing to prompt effectiveness but also demonstrates significant improvements in experimental results.

**Reasons To Reject:**

1. The absence of a complexity analysis is notable. High computational complexity could significantly hinder practical applicability.

2. Moreover, the reliance on a single language model as the backbone is limiting, potentially confining the proposed model's effectiveness solely to GPT-2 large, reducing the persuasiveness of experimental outcomes.

3. Additionally, the limited baseline, restricted to AutoPrompt, further weakens the paper's credibility.

4. No code and data provided， lacking of reproducibility of experimental results.

With these concerns in mind, I kindly suggest revisiting and addressing these issues comprehensively to enhance the overall quality and rigor of the paper before reconsidering it for acceptance. Your efforts in refining these aspects would undoubtedly contribute to a more robust and impactful submission.

**Reproducibility:**

2: Would be hard pressed to reproduce the results. The contribution depends on data that are simply not available outside the author's institution or consortium; not enough details are provided.

**Reviewer Confidence:**

3: Pretty sure, but there's a chance I missed something. Although I have a good feel for this area in general, I did not carefully check the paper's details, e.g., the math, experimental design, or novelty.

**Typos Grammar Style And Presentation Improvements:**

line 028: "langauge" -> "language"
line 271-272: "langauge" -> "language"

---

> ### Author Rebuttal · Authors · 2023-08-29
>
> Due to the number of reviews we got for this work, we have to prioritize some common questions raised in all reviews and make a general response to all reviewers. However, we will be super happy to discuss and follow up here if you think any of your crucial concerns are not addressed. Thank you so much! We additionally thank your suggestions on presentation improvements.

---

### Meta-Review · Area_Chair_8RAu · 2023-09-20

**Recommendation:** 4

**Metareview:**

The question paper rotates around the automatic prompt generation theme for NLP tasks. The authors delve deep into the mechanics of prompt tuning, emphasizing human readability and the efficiency of the prompts. Their primary contributions include introducing the FluentPrompt method, analyzing factors making prompts effective, and an unsupervised approach to generating prompts without relying on labeled data. While there is consensus on the paper's merits, some reviewers have raised concerns about certain study factors.

**Reason To Accept**
- Originality: The paper offers the NLP community fresh insights through several points, which detail the attributes of a good prompt. The characteristics emphasized—topic relevance and calibration for label bias help understand why prompt tuning is practical.
- Methodological Contributions: The proposed FluentPrompt technique is an innovative solution incorporating fluency constraints to generate effective prompts. Along with an unsupervised approach, this method consistently outperformed baselines across experiments.
- Analysis: Reviewers praised the exploration of factors contributing to prompt efficacy. The emphasis on topic relevance and calibration aids researchers in understanding the nuances of effective, prompt design.
- Potential for Follow-up Research: The analyses presented, especially concerning prompts' attributes, are anticipated to inspire subsequent studies in the domain.
**Reason To Reject**
- Limitations: Despite its potential, the FluentPrompt method is not perfect. The so-called 'human-readable' prompts sometimes need to be clarified, raising concerns about their true interpretability.
Scope of Analysis: Several reviewers felt the analysis need to be narrower in focus. The effectiveness of prompts is primarily studied for classification problems, which might only generalize to some NLP tasks.
- Comparison with Baselines: The paper mainly contrasts FluentPrompt with only a few baselines, such as Empty Prompt and AutoPrompt. Some reviewers wanted a broader comparison, including methods like continuous soft prompt tuning and other discrete prompt techniques.
- Exps Limitations: GPT-2 for experiments raises concerns about the findings' applicability across different models. The studys generalizability remains in question, as no other models were considered.
$FFinal Verdict:

While the paper introduces some innovative ideas and methods in the domain of prompt generation and tuning, certain reservations exist about its breadth and depth of analysis. Addressing these concerns might enhance its credibility and value to the NLP community.




**Overall Summary**
While the paper introduces some innovative ideas and methods in the domain of prompt generation and tuning, certain reservations exist about its breadth and depth of analysis. Addressing these concerns might enhance its credibility and value to the NLP community.

---

### Decision · Program_Chairs · 2023-10-07

**Decision:**

Accept-Findings

**Comment:**

The question paper rotates around the automatic prompt generation theme for NLP tasks. The authors delve deep into the mechanics of prompt tuning, emphasizing human readability and the efficiency of the prompts. Their primary contributions include introducing the FluentPrompt method, analyzing factors making prompts effective, and an unsupervised approach to generating prompts without relying on labeled data. While there is consensus on the paper's merits, some reviewers have raised concerns about certain study factors.

**Reason To Accept**
- Originality: The paper offers the NLP community fresh insights through several points, which detail the attributes of a good prompt. The characteristics emphasized—topic relevance and calibration for label bias help understand why prompt tuning is practical.
- Methodological Contributions: The proposed FluentPrompt technique is an innovative solution incorporating fluency constraints to generate effective prompts. Along with an unsupervised approach, this method consistently outperformed baselines across experiments.
- Analysis: Reviewers praised the exploration of factors contributing to prompt efficacy. The emphasis on topic relevance and calibration aids researchers in understanding the nuances of effective, prompt design.
- Potential for Follow-up Research: The analyses presented, especially concerning prompts' attributes, are anticipated to inspire subsequent studies in the domain.
**Reason To Reject**
- Limitations: Despite its potential, the FluentPrompt method is not perfect. The so-called 'human-readable' prompts sometimes need to be clarified, raising concerns about their true interpretability.
Scope of Analysis: Several reviewers felt the analysis need to be narrower in focus. The effectiveness of prompts is primarily studied for classification problems, which might only generalize to some NLP tasks.
- Comparison with Baselines: The paper mainly contrasts FluentPrompt with only a few baselines, such as Empty Prompt and AutoPrompt. Some reviewers wanted a broader comparison, including methods like continuous soft prompt tuning and other discrete prompt techniques.
- Exps Limitations: GPT-2 for experiments raises concerns about the findings' applicability across different models. The studys generalizability remains in question, as no other models were considered.
$FFinal Verdict:

While the paper introduces some innovative ideas and methods in the domain of prompt generation and tuning, certain reservations exist about its breadth and depth of analysis. Addressing these concerns might enhance its credibility and value to the NLP community.




**Overall Summary**
While the paper introduces some innovative ideas and methods in the domain of prompt generation and tuning, certain reservations exist about its breadth and depth of analysis. Addressing these concerns might enhance its credibility and value to the NLP community.